# From Microenvironment Remediation to Novel Anti-Cancer Strategy: The Emergence of Zero Valent Iron Nanoparticles

**DOI:** 10.3390/pharmaceutics14010099

**Published:** 2022-01-02

**Authors:** Ya-Na Wu, Li-Xing Yang, Pei-Wen Wang, Filip Braet, Dar-Bin Shieh

**Affiliations:** 1School of Dentistry & Institute of Oral Medicine, National Cheng Kung University Hospital, National Cheng Kung University, Tainan 701401, Taiwan; yana.wu@gmail.com (Y.-N.W.); peiwen1005@gmail.com (P.-W.W.); 2The i-MANI Center of the National Core Facility for Biopharmaceuticals, Ministry of Science and Technology, Taipei 10622, Taiwan; 3Department of Photonics, National Cheng Kung University, Tainan 70101, Taiwan; fingerzoo@gmail.com; 4Australian Centre for Microscopy & Microanalysis, The University of Sydney, Sydney, NSW 2006, Australia; filip.braet@sydney.edu.au; 5Faculty of Medicine and Health, School of Medical Sciences (Discipline of Anatomy and Histology), The University of Sydney, Sydney, NSW 2006, Australia; 6Charles Perkins Centre (Cellular Imaging Facility), The University of Sydney, Sydney, NSW 2006, Australia; 7Center of Applied Nanomedicine, National Cheng Kung University, Tainan 701401, Taiwan; 8Core Facility Center, National Cheng Kung University, Tainan 701401, Taiwan; 9Department of Stomatology, National Cheng Kung University Hospital, Tainan 704302, Taiwan

**Keywords:** zero-valent iron, cancer, ferroptosis, macrophages, angiogenesis, nanomedicine, nanoparticles, ROS, tumor, tumor microenvironment

## Abstract

Accumulated studies indicate that zero-valent iron (ZVI) nanoparticles demonstrate endogenous cancer-selective cytotoxicity, without any external electric field, lights, or energy, while sparing healthy non-cancerous cells in vitro and in vivo. The anti-cancer activity of ZVI-based nanoparticles was anti-proportional to the oxidative status of the materials, which indicates that the elemental iron is crucial for the observed cancer selectivity. In this thematic article, distinctive endogenous anti-cancer mechanisms of ZVI-related nanomaterials at the cellular and molecular levels are reviewed, including the related gene modulating profile in vitro and in vivo. From a material science perspective, the underlying mechanisms are also analyzed. In summary, ZVI-based nanomaterials demonstrated prominent potential in precision medicine to modulate both programmed cell death of cancer cells, as well as the tumor microenvironment. We believe that this will inspire advanced anti-cancer therapy in the future.

## 1. Introduction

According to GLOBOCAN 2020, cancer is one of the leading chronic diseases causing death before the age of 70 years in many countries [1]. Almost 10 million cancer deaths occurred in 2020 [1]. However, developing effective anti-cancer therapeutics is still a clinically significant unmet need. Recent advancements in the translational applications of nanomaterials in clinical settings bring tremendous opportunities for solving such critical clinical unmet needs through enabling efficient early disease diagnostics and therapeutics. For advanced disease stages, cancers are mainly treated by chemotherapy, radiotherapy alone for non-resectable cases, or in combination with surgery. However, most traditional anti-cancer compounds are cytotoxic to actively dividing normal cells in the patients, which renders significant clinical side-effects and complications. To improve selectivity toward cancerous cells, targeting therapies based on molecular signatures (mainly specific oncogenic mutations or aberrant expressions) were developed as small molecular compounds or macromolecules. Unfortunately, undesirable symptoms were also reported. For example, the humanized monoclonal antibody trastuzumab was approved to treat HER2-positive breast cancer, via binding to HER2 domain IV, which is associated with considerable cardiotoxicity [2]. However, cancer cells that escape from targeting therapy develop resistance due to hyperactivation of the mTOR pathway and a metabolic shift to glycolysis [3]. The major goal is to improve therapeutic activity and selectivity in the development of anticancer agents.

Reactive oxygen species (ROS), defined as unstable and reactive molecules containing oxygen, are normal byproducts and signaling messengers of extensive cellular processes, including oxygen metabolism [4]. A moderate level of ROS is essential for various physiological processes, including the cellular signaling that regulates cell survival and proliferation [5]. ROS also played an important role in cancer development [6], as well as various pathological processes [7]. Due to the nature of uncontrolled hyperproliferation, it has been reported that cancer cells demonstrate a higher basal ROS level than normal cells [8]. Since adaptation to conditions of excessive ROS is necessary for cancer cells, a higher level of antioxidative capacity in cancer cells has been reported in several studies [9]. Intriguingly, the redox systems of cancer cells are more vulnerable than those of normal cells [5,10]. Therefore, a serial of exogenous ROS generating agents may increase the ROS level over the cytotoxic threshold and thus achieve the anti-neoplastic goal selectively [11,12].

Over the past two decades, nanomedicine made immense progress in developing smart solutions to deliver and target anti-cancer drug formulations [13]. Pharmacokinetics and pharmacodynamics are critical to accumulate sufficient therapeutic doses in cancer lesions while minimizing their retention in major organs. Nanomaterials, such as VYXEOS^®^ or BIND-014^®^, were designed to carry anti-cancer compounds alone or with active targeting to achieve targeted drug delivery [14,15,16]. Nanomaterials mainly serve as the biocompatible bioinert ingredients in therapeutic designs [3]. On the emergence of nanotechnology, other highly innovative nanomedicine solutions have been developed, such as the relay of externally applied energy, mainly electromagnetic or acoustic waves, and brought promising directions for future anti-cancer strategies [17,18], including advanced radio-photodynamic therapy [19], chemodynamic therapy [20,21], sonodynamic therapy [22], and etc. In 2008, that nanoparticles (NPs) that possess the strong endogenous anti-cancer activity and selectively suppress cancer cells were reported [23]. Despite metal oxide nanoparticles being widely used in therapeutic applications (e.g., zinc, iron, titanium, copper, etc.), zerovalent iron (ZVI) NPs exhibited distinct anti-cancer properties to modulate solid tumors in comprehensive approaches.

The ZVI nanoparticle was originally developed with the intent to improve the magnetic properties of iron oxide-based NPs, which have long been of particular interest in magnetic resonance imaging (MRI) contrast agents. Various iron oxide NPs have been approved by the FDA as T2-negative contrast agents for MRI and further extended to the alternating magnetic field induced hyperthermotherapy, or combined theranostics [14,24]. Interestingly, ZVI NPs have been extensively applied for the reduction of various organic or inorganic pollutants in wastewater treatment and environmental remediation [25,26], through generation of massive ROS via Fenton reactions and other chemical processes. Different from iron oxide NPs, ZVI NPs also generate a much stronger Fenton reaction as many metal-based nanomaterials in the cells [27,28,29]; their products, called iron oxides, are generally recognized to be biocompatible and could be excreted from the body through sophisticated iron ion transportation systems [30,31]. 

Accumulated evidence showed that ZVI NPs generated significantly higher ROS in cancerous cells, loss of mitochondria membrane potential associated with distinct lysosomal activity and intracellular abundance, altered mitochondria structure, and energy metabolism. Therefore, an intrinsic property of ZVI NPs was to selectively eliminate cancer cells while being less harmful for healthy cells in vitro and in vivo. ROS was recently described as an important signal transduction mediator in various cellular functions and cell types. Iron oxides NPs were recently reported to modulate tumor immunity through reprograming their differentiation and affecting anti-cancer phenotypes [32]. Nevertheless, the multiple roles of ZVI NPs in the modulation of both cancer cell death and stemness —as well as angiogenesis and immune response in the tumor microenvironment (TME)— have not been comprehensively reviewed or discussed. This manuscript aims to provide an in-depth overview of the latest achievements in ZVI NPs fabrication and reviews the intrinsic anti-cancer activity of ZVI-based NPs reported in the literature. The underlying mechanisms will be addressed, from the molecular to the systemic level.

## 2. Synthesis and Characterization of ZVI NPs

### 2.1. Synthetic Approaches of ZVI NPs

The ferrous ZVI NPs can be produced using physical or chemical methods, either by top-down or bottom-up approaches [33,34]. The bottom-up approach involves assembling individual atoms or molecules to form nano-sized structures. It uses a wide range of reductants, to convert dissolved iron ions in the solution to ZVI, and to grow into NPs. The top-down approach, on the contrary, involves the crushing of bulk particles (microscale or granular) of iron to fine nano-sized particles, by mechanical or chemical methods. Properties of ZVI NPs strongly depend on their size, surface, capping materials, oxide layer, or sub-particle architectures. Bare ZVI NPs are highly reactive [33] and can easily agglomerate due to their high surface energy and magnetic properties. When in contact with air or water, the surface of ZVI NPs is covered with a thin oxide layer, which appears to be a mixed Fe(0)/Fe(II)/Fe(III) phase, with an apparent surface close to FeOOH [35]. It is believed that the major phase is lepidocrocite (-FeOOH) [36]. A core-shell ZVI NPs with the oxide shell can be advantageous in practical applications, due to their higher stability than bared pyrophoric ZVI NPs. Agglomeration of ZVI NPs can be improved by modifying the surface, using organic polymers and polyelectrolytes simultaneously, or in a subsequent step of the synthesis process. The methods commonly used in ZVI NPs production will be discussed in the following sections.

#### 2.1.1. The Top-Down Synthesis

Milling processes are the frequently used methods for the top-down approach to producing ZVI NPs. The iron materials (e.g., carbonyl iron, cast iron, sponge iron, and iron powder) are milled to fine nano-sized particles (e.g., ball-mills, vibrating mills, and stirred ball mills) under inert environments. The advantage of this method is that it is relatively cheaper than using chemical reductants and easy to scale up. It is widely used in the industry for large scale production [37]. Milling in an inert atmosphere produces very reactive and pyrophoric iron particles, which ignite spontaneously upon contact with air. To reduce the burn hazard and reactivity of ZVI NPs, a capping layer is usually developed by performing milling in a grinding media. To solve stability and safety issues, several grinding media and additives were developed. The grinding media containing water can aid in the formation of a stabilizing oxide shell over the NPs, but corrosion inhibitors and surfactants were also used to help in the fabrication of ZVI NPs [38,39,40]. Laser ablation is another method to synthesize Fe NPs on a small scale. Irradiation of an iron metal target with a laser pulse locally melts and vaporizes the metal. Hot metal atoms are cooled by the surrounding medium to form metal NPs [41]. Noble gas sputtering is rarely used for ZVI NPs synthesis. When iron is used as the sputtering target, a super-saturated iron vapor forms during the sputtering process, which can be condensed into solid clusters by the subsequent cooling process [42].

#### 2.1.2. The Bottom-Up Synthesis

The most used method for ZVI NPs synthesis in laboratories is the borohydride route, where iron salts are reduced by sodium borohydride in an aqueous environment under an inert atmosphere (Equations (1) and (2)). The typical diameter of NPs obtained in this process is generally below 100 nm. Another synthetic route through a reverse-micelle (water-in-oil microemulsions) system can generate much smaller (~10 nm) and controllable particle sizes than the aqueous solution approach [43,44], but the complex preparation and synthesis process limits its application to large-scale production. The reductive reactivity of ZVI is sensitive to the initial concentration of the iron precursor and the loading ratio of borohydride to ferric ions. A higher rate of adding NaBH_4_ BH_4_^−^: Fe^2+^ (up to 1:3) can yield higher reactivity for the synthesized ZVI NPs [45,46]. Ultrasound sonication during the synthesis strongly influenced particle size and shape. Under higher ultrasonic power, the morphology of NPs changed from spherical to plate and needle shapes. The particle size also decreased when higher ultrasonic power and precursor/reductant ratios were used [47,48].
2 Fe^2+^_(aq)_ + BH_4_^−^ + 3 H_2_O → 2 Fe^0^_(s)_ + H_2_BO_3_^−^ + 4 H^+^_(aq)_ + 2 H_2(g)_
(1)
4 Fe^3+^_(aq)_ + 3 BH_4_^−^ + 9 H_2_O → 4 Fe^0^_(s)_ + 3 H_2_BO_3_^−^ + 12 H^+^_(aq)_ + 6 H_2(g)_(2)

The borohydride synthetic route is very popular in laboratories for producing ZVI NPs on a small scale due to its simplicity, as no special instruments or materials are needed. In addition, the products have a homogenous structure and are very reactive [49]. The borohydride route is also widely used for synthesizing capped and supported ZVI-based nanomaterials to minimize aggregation, increase stability and mobility in soil remediation, and reduce leaching of ZVI NPs. A wide range of inorganic and organic support and capping materials have been used, including metal oxides, silica, carbon materials, monomers, surfactants, or polymers, to provide hydrophobic, hydrophilic, or amphiphilic surface properties for ZVI NPs [7,33,50,51]. The synthesis of inorganic ZVI NP carrier materials is usually based on the chemical process where carriers are added to the iron ion solution before NaBH_4_. The chemical reduction method is complex and includes multiple steps such as preparation of the degassed solution, nucleation of the ZVI cluster, growth of ZVI nuclei, and agglomeration of ZVI NPs, as well as washing, separation, and dehydration of the NPs. These steps may lead to the formation of a thin capping oxide layer on the iron surface, depending on the conditions of synthesis [52]. Other reductants such as hydrazine [53], sodium dithionite [54], or green synthesis methods, using plant extracts [7,55,56] and microbial extracts [57,58,59] as the reductants, can also successfully produce ZVI NPs. 

The thermal decomposition of iron pentacarbonyl [Fe(CO)_5_] in a high boiling point solvent generates small and uniform sized ZVI NPs (about 10 nm in diameter) [60,61], but iron pentacarbonyl is highly toxic and unstable, thus such an approach raises critical safety concerns. Another well-known commonly used approach, for producing high-quality particles with more narrow size distribution, is through thermal decomposition of an iron precursor. The iron precursor (iron oxide or an iron salt) is reduced at high temperatures (>500 °C) in the presence of gaseous reducing agents such as H_2_, CO_2_, or CO [62]. The precursor, capping agent ratio, rate of heating, the final reaction temperature, and the annealing time are all important factors in controlling both the diameter and the size distribution of the nanoparticles produced [63]. 

Electrochemical deposition of iron from solutions is a low-cost and high-efficient method to synthesize ZVI nanomaterials. Particles synthesized in this way are usually larger than the nanometer domain with only one nano-dimension [64]. However, combining electrochemical and ultrasonic techniques—thereby removing ZVI NPs instantaneously from the cathode into the solution, including the use of surfactants to stabilize NPs—could produce NPs with a diameter of 1–20 nm [65].

### 2.2. The Coating and the Storage of ZVI NPs

The non-oxidized ZVI elements were found to be the active ingredient for the observed selective anti-cancer effect of ZVI-based NPs [43]. Wu et al. reported that ZVI-based NPs with gold shells gradually lost their anti-cancer activity when stored or “aged” in water or air [43,66]. The “aged” NPs could be decomposed into separated clusters of irregular fragments of γ-Fe_3_O_4_ and agglomerated gold clumps. However, the anti-cancer property of iron core-gold shell NPs can be well preserved for at least one year when the NPs are stored under argon gas. It could be one of the reasons why the reported anti-cancer activities of ZVI-based NPs varies in the literature. It was reported by Liang et al. that the half maximal inhibitory concentration (IC_50_) of serial porous yolk-shell Fe/Fe_3_O_4_ NPs dispersed in buffer was slightly increased after one-week of air exposure [67]. According to the research from Yang et al., the oxygen-to-iron ratio of ZVI-based NPs is highly correlated to their cytotoxic potency against cancer cells [68], as cytotoxicity can be compromised with the oxidation status of NPs. 

Furthermore, different types of coatings on ZVI-based NPs seem to be pivotal for the observed cytotoxic effects on cancer cells. According to the study by Yang et al. [69], mesoporous silica-coated ZVI (ZVI@mSiO_2_) NPs showed better cytotoxicity against cancer cells than that of the ZVI NPs fully covered by silica (ZVI@SiO_2_). Intriguingly, ZVI@SiO_2_ remained intact after internalization, whereas the mesoporous coated ZVI-based NPs deformed into a hollow and void structure after 24 h treatment in cancer cells. The oxidation process of ZVI-based NPS, as well as their decomposition into iron ions during lysosomal acidification, are crucial to trigger cancer-selective cytotoxicity. Appropriate surface coating design, as well as storage conditions, are critical for optimizing anti-cancer efficacy. Although ZVI confined mesoporous silica nanocarriers were designed for contrast imaging purposes, it is of note that such NPs showed limited cancer-selective cytotoxicity compared to other ZVI-based NPs [70]. The small doping of 3 nm ZVI NPs within mesoporous silica may contribute to a larger surface to volume ratio, and thus boost the oxidizing tendency of the ZVI elements. Generally, ZVI NPs coated with gold [43,66,71], silver [68,72,73], carboxymethyl cellulose (CMC) [74], or different types of silica [69,70] have all been reported to harbor significant cancer-selective cytotoxic effects, and could potentially be developed as effective theranostic anti-cancer agents, thereby combining MRI imaging and nanomedicine drug treatment regimens (Table 1). To summarize, the purpose of the optimal surface coating of the effective ZVI-based NPs is to stabilize and preserve the dispersity of the particles without compromising the release of iron ions within an acidic neoplastic environment. Compared to inorganic surface coating, the biocompatible organic surface coating, i.e., CMC, can be an optimal option when considering clinical practicality, especially as CMC has been approved by the U.S. Food and Drug Administration [75]. It is widely used in oral, ophthalmic, injectable, and topical pharmaceutical formulations. For future studies, the stability and biocompatibility of their coatings within biological media or body fluids should be seriously taken into consideration, and this must be addressed prior to future clinical studies. 

## 3. Mechanisms of the Endogenous Anti-Cancer Efficacy

### 3.1. Tumors and Their Microenvironment 

Tumorigenesis is a pathological process involving the dysregulation of oncogenes and tumor suppressor genes during the initiation and progression of the disease. It is well known that cancer cells are characterized by self-sufficiency in growth signals, insensitivity to anti-growth signals, tissue invasion and metastasis, limitless replicative potential, sustained angiogenesis (blood vessel growth), and evasion of cell death [78]. Recent studies showed that two new hallmarks were added to the list, namely avoiding immune destruction and reprogramming energy metabolism [78]. Tumor cells undergoing hyperproliferation usually rely on reprogrammed metabolism and are mostly accompanied by abundant ROS production. High oxidative stress is toxic to cells when it exceeds tolerable levels, and may trigger senescence, apoptosis, ferroptosis, or other types of programmed cell death. Cells developed a sophisticated mechanism to sense and manage stress. Despite cancer cells often being under persistent oxidative stress [79,80], this stress is insufficient to cause cell death. The dedicated oxidative balancing machinery enabled not only the survival of the tumor cells, but also made them highly chemo-resistant. Glutathione (GSH) and thioredoxin systems are the two major antioxidant systems in the cells. There is accumulated evidence that showed that GSH and thioredoxin synergistically promote carcinogenesis [81,82]. Inhibition of the two systems in cancer cells has been developed as a new cancer therapeutic strategy [83]. To mitigate the ROS burden, a glycolytic shift in cancer cells was associated with carcinogenesis initiation and tumor progression [84]. Hypoxia in the TME further boosts glucose uptake and glycolytic activity. The glycolytic intermediates entering the pentose phosphate pathway result in higher nicotinamide adenine dinucleotide phosphate (NADPH) levels, which played a pivotal role in reducing oxidative stress. Such metabolic reprogramming activates the master antioxidant machinery through a key regulator, the nuclear factor erythroid 2–related factor 2 (NRF2) [84]. The role of the transcription factor Nrf2 in carcinogenesis is controversial. In many cancers, the Nrf2-mediated transcription system is continuously activated through somatic mutations, induces various downstream antioxidant systems [85], and confers resistance to chemotherapy [86]. Suppression of Nrf2 in these cancers may be beneficial for clinical therapy.

Accumulated evidence showed that interaction between cancer cells and cellular components within the TME significantly affects patient clinical prognosis. These include immune cells, fibroblasts, endothelial cells, and other types of stromal cells [87]. The tumor-associated macrophages could be classified into two groups: the pro-inflammation or anti-cancer M1, and the immunosuppressive or pro-tumor M2. Tumor-infiltrating T cells directly interact with M2 cells through affinity binding of programmed cell death ligand 1 (PD-L1) on the cancer cell membrane and the programmed cell death protein 1 (PD-1) on M2 cells. Such interaction leads to T-cell anergy and immune escape for tumor cells. Macrophage polarization is closely associated with differential regulation of the iron metabolism [88]. M1 macrophages favor iron accumulation and show a Ferritin^high^ Ferroportin^low^ phenotype. In contrast, the iron content of the anti-inflammatory M2 macrophages is lower and presents a Ferritin^low^ Ferroportin^high^ phenotype [89]. Exposure to iron oxide nanoparticles within the TME could lead to polarization of the pro-tumoral M2 to anti-tumoral M1 macrophages [90], which inhibit tumor growth and metastasis in vivo [91]. It is believed that iron oxide NPs modulate only the tumor-associated macrophages in the TME, while not significantly affecting cancer cells nor angiogenesis. 

### 3.2. The Therapeutic Implication of ZVI 

Interestingly, while ZVI-based NPs showed prominent functions in eliminating pollutants in the ecosystem on earth [92], they also demonstrated great potential in the remediation of the microenvironment inside and outside cancer cells. Different research groups have reported that various types of ZVI-based NPs have demonstrated inherent anti-cancer efficacy with high selectivity to cancerous cells while spared healthy normal cells both in vitro and in vivo (Table 1). Such inherent cancer-selective cytotoxicity was achieved without externally applied energy or carrying anti-cancer drugs. Accumulated evidence showed that iron ions converted and released from ZVI-based NPs played critical roles in such selectivity, to block multiple oncogenic signaling cascades within the cancer cells and the TME. At molecular, organelle, cellular, and system biologic levels, ZVI-based anti-cancer therapy has just recently been dissected and revealed (Table 1).

One of the pioneer observations about the distinct properties that ZVI-based NPs exhibited to selectively inhibit cancer cell proliferation while sparing normal cells was reported by Wu et al. [43]. In the report, iron core gold shell NPs (Fe@Au) were synthesized using the reverse emulsion method. The Fe@Au selectively and significantly inhibited proliferation of oral- and colorectal-cancer cells in vitro at doses as low as 5 μg/mL, but had a small adverse effect on normal healthy control cells [43]. Without significant difference in the NP uptake between cancer and control cells, Fe@Au caused an irreversible membrane-potential loss in the mitochondria of cancer cells, but only a transitory decrease in membrane potential in healthy control cells. Iron elements, before oxidation, triggered mitochondria-mediated autophagy was identified as the key factor responsible for the differential cytotoxicity observed between cancerous and healthy cells [71]. In the in vivo study, Fe@Au intratumoral injection was able to reduce tumor volume to 40% of the control group. 

In 2016, Zhang et al. [77] further reported that ZVI NPs comprising amorphous iron were applied in the treatment of human breast adenocarcinoma. He discovered that ZVI triggered a localized Fenton reaction inside cancer cells [77]. A rapid ionization of the amorphous iron NPs under an acidic cancer microenvironment enables ferrous ions to release, which further reacts with hydrogen peroxides in tumor regions. Through the Fenton reaction, hydroxyl radicals were extensively generated leading to the observed anti-cancer efficacy in vitro and in vivo. The IC_50_ of amorphous iron NPs is about 100 µg/mL in the in vitro study under the presence of H_2_O_2_ (50 µM). At treatment doses of 15 mg/kg via intratumoral injections, it led to complete inhibition of tumor growth within 16 days, while intravenous injection of 75 mg/kg of the ZVI NPs reduced tumor volume to about 80% of the control group. Interestingly, the amorphous iron NPs were not able to kill cancers without hydrogen peroxide and an acidic environment (IC_50_ > 200 µg/mL) in the in vitro model. It was suggested that the anti-cancer effect of amorphous ZVI depends on the ionization of amorphous ZVI NPs for release of ferrous ions and subsequent H_2_O_2_ disproportionation for efficient hydroxyl radical generation. 

Shevtsov et al. reported that ZVI confined within mesoporous silica nanocarriers (Fe(0)@MCM-41) resulted in significantly enhanced cytotoxicity in several different tumor cells and normal cells compared to the carrier alone. The cytotoxicity was exerted through promoting ROS production via the ZVI component [70]. While Shevtsov showed that Fe(0)@MCM-41 failed to present selective cytotoxicity toward cancerous cells in their rat glioma, glioblastoma, leukemia, or adenocarcinoma cells versus splenocytes and fibroblasts, Yang et al. [69] reported a contradictory result, whereby ZVI@mSiO_2_ exhibited a cancer-selective inhibition in human oral cancer and paired oral keratinocytes in vitro and in vivo. In Shevtsov’s study, the IC_50_s of the NPs to cancer versus normal fibroblasts and spherocytes are all within 50–150 µg/mL. On the other hand, Yang reported IC_50_ for oral carcinoma at 5 µg/mL, while it was greater than 50 µg/mL for the fibroblasts. One of the possible reasons is that the ZVI@mSiO_2_ NPs from Yang were less oxidative than those synthesized by Shevtsov. As acidification of lysosomes is more aggressive in cancerous cells, Fenton reaction and subsequent ROS generation are expected to be more extensive in cancer cells than in normal cells, rendering the differential response in Yang’s study. On the other hand, such differential response was not observed by Shevtsov [69]. 

In 2019, Anbouhi et al. [74] synthesized ZVI NPs by reducing ferrous ions of FeSO4⋅7H2O by sodium borohydride in an ethanol-water mixture. The derived 37 nm ZVI NPs were able to induce human neuroblastoma cell death in a dose-dependent manner with an IC_50_ of 47.9 µg/mL, while being biocompatible with human white blood cells [74]. The author further reported that the NPs could bind to human serum albumin through hydrogen bonds and van der Waals interactions, therefore this might affect their pharmacokinetics.

However, it was not until Huang’s report that detailed molecular genetic mechanisms responsible for ZVI-triggered selective anti-cancer cell activity were disclosed. Huang is also the first to demonstrate that resistance to nanoparticles, particularly ZVI-based nanoparticles, could be developed in cancer cells. In his recent report using microarray analysis of paired ZVI NPs sensitive/resistance cells, he showed that ferroptosis is the common and key mechanism for the derived cancer-selective cytotoxicity by a wide variety of ZVI-based NPs, including bare ZVI, ZVI@CMC, and CMC stabilized ZVI NPs. Different from previous reports, ZVI-based NPs triggered lipid peroxidation specifically in mitochondria, rendering their loss of membrane potential and dysfunction. ZVI NPs also reduced glutathione peroxidases (GPX4) level, thus lowered detoxification capacity of ROS, blocked NADPH supply, and enhanced their sensitivity to ferroptosis inducers. As ferroptosis signaling played a key role in ZVI-resistant cancer cells, Huang showed that co-treatment of ferroptosis inducers could re-sensitize the ZVI-resistant cells in vitro and in vivo without compromising healthy normal cells, nor causing weight loss in vivo by co-treatment of ferroptosis inducers.

Yang et al. also demonstrated a positive correlation between the oxidative index (oxygen-to-iron ratio) and IC_50_ in the oral cancer cells treated with various ZVI-based nanoparticles [68]—the less the oxidative status of ZVI, the higher the anti-cancer capacity. Furthermore, Yang also demonstrated that aged ZVI NPs had increased oxygen composition and reduced cytotoxicity, as well as attenuated ROS. These findings suggested that the maintenance of the non-oxidized ZVI components is important for their cancer-killing ability. The oxidation of ZVI and iron ion release is another key step for harboring the cytotoxicity of ZVI NPs. In 2019, Yang and colleagues also reported that only ZVI@mSiO_2_ with mesoporous pores coverage on ZVI NPs showed significant cytotoxicity to cancer cells, including concurrent void structure after cancer cell uptake. However, ZVI@SiO_2_ covered with intact SiO_2_ showed no cytotoxicity and intact structure in TEM cell sections [69]. This provides another explanation for ZVI NPs digested by cells to release iron and trigger downstream oxidative stress to kill cancer cells. In 2020, Hashemi et al. synthesized ZVI NPs by *Feijoa sellowiana* fruit extract displayed not only the antibacterial but also anti-cancer activities against two tumor cell lines with little effect on normal cells [73]. Liang et al. proposed in 2021 a strategy to deliver ZVI via a porous yolk-shell nanostructure with superior tumor inhibition activity in hepatoma cells [67]. Despite the different synthetic processes, the synthesized ZVI NPs still possessed similar inherent anti-cancer activity.

In 2021, Hsieh et al. proposed a systemic observation on the mechanisms of ZVI triggered cancer-selective cytotoxicity in lung cancer [72]. The authors proposed that ZVI not only caused mitochondria dysfunction, ROS production, ferroptosis, but also that this type of ZVI-based NPs attenuated the self-renewal ability of cancer and downregulated angiogenesis-related genes. The degradation of NRF2 triggered by ZVI-based NPs exposure further caused redox unbalancing and triggered ferroptosis cell death via excessive oxidative stress and lipid peroxidation. Intriguingly, ZVI NPs also augmented anti-cancer immunity via shifting pro-tumor M2 macrophages to anti-tumor M1, minimizing the population of regulatory T cells, attenuating PD-L1 expression in cancer cells, downregulating PD-1 and cytotoxic T-lymphocyte associated protein 4 in CD8+ T cells, to potentiate their cytolytic activity against cancer cells. Such therapeutic potential in immunocompetent and humanized mice has also been validated. This study concluded that ZVI NPs have promising integrated anti-cancer modalities that suppress cancer growth and metastasis through multiple approaches and via a systemic manner. 

### 3.3. The Underlying Mechanisms

#### 3.3.1. Lysosomes and Acidic TME 

Lysosomes play a key role in the regulation of iron metabolism, they occupy a central position in iron homeostasis, and control cell-death signaling [93]. Lysosomes also play a pivotal role in the resistance of chemotherapy, by sequestering drugs in their acidic environment [94]. In malignant tumors, cancer cells shift their glucose metabolism from oxidative phosphorylation to lactate fermentation and produce massive protons to meet their anabolic demand. Thus, cancer cells may activate multiple acid removal pathways that result in the acidification of the extracellular microenvironment [95]. Vacuolar H^+^-ATPase (V-ATPase) is a proton pump that pumps protons into the lumen of the lysosome and creates the acidic pH in the organelles. V-ATPase also affects TME by proton extrusion into the extracellular space [96]. The V-ATPase has since been found to be overexpressed in numerous invasive cancer cell types [97], and some are correlated with poor patient survival. ZVI-based NPs take the advantage of the acidic nature of cancer and deliver and release massive iron ions specifically within tumoral regions via a “lysosome-enhanced trojan horse effect [98].” In a study by Yang et al., only the ZVI@mSiO_2_ successfully released massive iron ions within cancer cells specifically, whereas ZVI-base NPs with fully covered silica did not [69]. When cancer cells were treated with an endosome-lysosome selective acidification inhibitor, the iron ions releasing profile is significantly reversed in ZVI@mSiO_2_ treated cancer cells, similar to the status of ZVI completely covered by SiO_2_ (ZVI@SiO_2_) treated cells. The silica shell remained a void structure after iron ions were released from ZVI@mSiO_2_ as observed under the transmission electron microscope equipped with energy-dispersive X-ray spectroscopy. The massive amounts of iron ions within cells, together with an acidic environment, are perfect conditions for the Fenton reaction to occur and hence produce tremendous oxidative stress. Superoxide is known to facilitate iron mobilization from ferritin to increase the labile pool of catalytic ferrous ions [99]. As a result, the oxidative stress is magnified and ultimately tilts the balancing machinery within cancer cells. As such, ZVI-based NPs trigger the tumoral cytotoxic effects in response to the acidic nature of the tumor cells (Figure 1).

#### 3.3.2. ROS and Lipid Peroxidation

Regarding the intrinsic anti-cancer activity of ZVI-based NPs, the amount of ROS production after treatment by ZVI-based NPs was observed in most cases [43,68,69,70,71,72,74,77,100]. ROS is proportional to the freshness [68], dosage [72], and iron to oxygen ratio of the ZVI-based NPs. These parameters are closely correlated with the observed cytotoxic potency against the cancer cells. Deferoxamine, ciclopirox olamine, and vitamin E treatment can be used to reverse the ROS production as well as the cytotoxicity, implying the importance of ferric ions [68,70,72,74]. ROS production was compromised when the ZVI-based NPs were fully covered with silica, resulting in the loss of the anti-cancer activity for NPs [69]. It is also of note that downregulation of GSH peroxidase [72,74] and GSH concentration [100] could be triggered by ZVI and 2D MXenes NPs. The evidence suggested that ZVI-based NPs may selectively induce massive ROS in cancer cells selectively and also deplete GSH from the balanced redox system in cancer cells. Therefore, while boosting the massive ROS production within cancer cells, ZVI-based NPs also compromised the tolerance of cancer cells to oxidative stress at the same time, thus achieving cancer-selective cytotoxicity.

When free radicals attack lipids, they cause a chain reaction of oxidative degradation of lipids. Such lipid peroxidation might also functionally damage the membrane through induction of covalent modifications, which significantly alter its permeability and ultimately cause it to lose integrity [101,102]. For initiation of lipid peroxidation, the formation of a complex between iron and lipids is required [103]. By reaction with metals, lipid radical reactions can be reinitiated resulting in the propagation of radical reactions [104]. Lipid peroxides produced during the propagation phase can be reduced by GPX4 to lipid alcohols, or degrade into hydroxy fatty acids or reactive aldehydes, such as malondialdehyde and 4-hydroxynonenal [105,106]. Malondialdehyde is formed during oxidative degeneration as a product of free oxygen radicals, and this is an accepted indicator of lipid peroxidation [107]. It has been reported that ZVI-based NPs induced cytotoxicity was accompanied by mitochondrial lipid peroxides. A six-fold increase of malondialdehyde in the mitochondrial fraction was observed after treatment [74]. Furthermore, the intracellular level of 4-hydroxynonenal increased after ZVI NPs treatment. Of note, this increment was largely diminished by NRF2 overexpression, suggesting that ZVI NP-induced ferroptosis lipid peroxidation was predominantly caused by the inhibition of NRF2 [72]. 

#### 3.3.3. ZVI-Based NPs Induced Programmed Cell Death

Iron elements are crucial for fundamental cellular anabolic activities; however, ROS production-derived lipid-related ROS was highly associated with ferroptosis [93]. According to a previous study, gold-coated iron NPs-treated cancer cells underwent mitochondria-mediated autophagy [71]. The ROS-induced apoptosis was also observed morphologically in the amorphous ZVI NPs-treated cancer cells [77]. Ferroptosis in ZVI@CMC NPs [74], porous yolk-shell Fe/Fe_3_O_4_ NPs, and ZVI@Ag-treated cancer cells have also been reported. Both the necrosis and the apoptotic cell death was observed in ZVI@mSiO_2_ [69] and the bare ZVI NPs [76]. In another report, ZVI@Ag treated cancers underwent both apoptosis and autophagy at the same time [68]. Taken together, this suggests that ZVI-based NPs may trigger cell death under diverse mechanisms depending on their structure, components, and surface modifications—including types of cancer cells and types of their microenvironment. These all underpin the vigorous, yet diverse, anti-cancer activities of ZVI-based NPs in a wide range of malignant diseases. It has recently become clear that there are complicated molecular cross-talks between programmed cell death processes including apoptosis, necrosis, and autophagy [108]. Ferroptosis is believed to be a type of autophagy-dependent cell death [109]. The molecular signaling of programmed cell death is mediated through a more complicated approach than expected, and the underlying mechanisms through which ZVI-based NPs modulate cancer cell death required further systematic research to clarify in detail.

#### 3.3.4. Tumor Micro-Environment 

ZVI elements suppress cancer cell growth not only through inherent anti-cancer properties but also by modulating the TME. Hsieh et al. reported that ZVI NPs downregulated the expression of angiogenesis-related genes [72], including sonic hedgehog, TGF-β, and VEGF. ZVI-based NPs treatment resulted in prominent inhibition of endothelial cell growth, migration, and tubular formation. Interestingly, ZVI NPs treatment also shifted pro-tumor M2 macrophages to anti-tumor M1 and decreased the population of regulatory T cells in vitro and in vivo, within the tumor tissues derived from the treated cancer-bearing murine models with competent immune system. Hsieh further revealed macrophage polarization toward the anti-cancer M1 phenotype in cancer tissue, together with decreased the numbers of regulatory T cells. Such pro-anti-cancer immunity was also observed in the circulating blood. Interestingly, a decrease in PD-1 and cytotoxic T-lymphocyte associated protein 4 expression was evident in both tumor-infiltrating and circulating CD8+ T cells, while PD-L1 was downregulated by ZVI NPs treatment in cancer cells in vitro and in vivo. Taken together, ZVI NPs potentiated the cytolytic activity of immune cells within the TME to fight against cancer cells [72]. It has been reported that high ROS levels induced by iron overload can polarize macrophages to the M1 subtype through p53 acetylation [110]. Furthermore, a similar reprogram macrophage response was notified with iron oxide NPs treatment [32]. Since the acidity of the interstitium has resulted from their high metabolic activity and overexpression of V-ATPase, the pH in the TME is spatially and temporally heterogeneous compared to normal tissues. The attenuated immune responses are also highly associated with acidic TME [111]. This forms an ideal environment to decompose ZVI-NPs into massive ‘iron ion bombs’ specifically within the TME, and thus modulate the tumor-associated immune cells. Also, the newly formed tumor vessels in the TME are usually aberrant in form and architecture, which enables nano-sized drugs to leak preferentially into tumor tissues through the permeable tumor vessels and thus better deposits within the tumor. Such enhanced permeability and retention effect can deliver various nano-sized (100–400 nm) particles in about 2-fold concentration compared to normal organs [112], which also boosted the observed anti-cancer phenomenon triggered by ZVI-NPs treatment. Typically, within a stromal tumor, ZVI-NPs seem to eliminate cancer cells in a systemic and multi-dimensional manner, through modulation of the surrounding blood vessels and immune cells that compose the TME (Figure 2). 

### 3.4. The Opportunity of Precision Medicine in ZVI-Based Nanotherapy

Precision medicine has emerged as a new trend in clinical therapeutics, whereby health care providers offer and plan specific care for patients based on the person’s genes the underlying nature of the disease. Cancer is a dynamic disease, where a heterogeneous population of malignant cells with diverse molecular signatures may respond differentially to treatment. Therefore, companion diagnostics is often used to guide the decision-making of therapeutics to better predict clinical outcomes. Huang discovered that cancers could have a distinctively different response to nanoparticle treatment, particularly ZVI-based nanoparticles. Cancer cells could also develop resistance to ZVI NPs [74]. The ZVI-resistant cells attenuated the ZVI-induced oxidative stress and presented a different metabolic profile of mitochondrial functions than the sensitive cancers. Genetic analysis further showed that they had enhanced NADPH supply, reduced sensitivity against ferroptosis inducers, and better antioxidant capability. To predict their sensitivity to ZVI NPs treatment, six upregulated genes (GSR, AKR1C1, AKR1C3, AKR1B1, AKR1B10, and KYNU) and three downregulated genes (ACSL4, ZEB1, and NNMT) were reported in refractory compared to sensitive cells. Such panel of genes may be further developed as a guideline for the inclusion criteria of ZVI NPs-treatable cases. Interestingly, Huang also demonstrated that ZVI-refractory cancer cells can be re-sensitized by co-treatment of ZVI NPs with small molecular ferroptosis-inducers in his in vitro and in vivo studies, while not significantly affecting normal healthy cells or the bodyweight of the tumor-bearing mice. Despite the prominent effects of ferroptosis inducers, only limited compounds were successful in clinical trials and this is likely attributed to its lower potency and metabolic instability [113]. We proposed that, with the combination of ZVI-based NPs, ferroptosis inducers might have a new opportunity to gain their therapeutic applications through enhanced cancer lesion delivery by EPR effects, as well as by their synergistic activities. In Hsieh’s study, more candidate biomarkers were revealed to be responsive to ZVI-based nanoparticle treatment. Hsieh discovered that ZVI was able to downregulate the expression level of cancer stemness genes (octamer-binding transcription factor 4, Nanog homeobox, and SRY-Box transcription factor 2), pro-angiogenesis genes (Sonic hedgehog, transforming growth factor beta, and vascular endothelial growth factor), GPX4, and the protein expression levels of NRF2 (Figure 3). Furthermore, NRF2 targeting antioxidant genes, ROS detoxification genes (aldo-keto reductase family 1 member B, C1, C2, and C3), NADPH-production enzymes (isocitrate dehydrogenase (NADP(+)) 1, malic enzyme 1, and phosphogluconate dehydrogenase), NADPH-dependent enzymes (NADH:Ubiquinone oxidoreductase complex assembly factor 4 and apoptosis inducing factor mitochondria associated 2) were decreased after ZVI based NPs treatment in vitro. Besides, mRNA expression levels of NRF2 target genes including solute carrier family 7 member 11, GPX4, solute carrier family 40 member 1, and aldo-keto reductase family genes were suppressed in ZVI-based NPs treated xenografts in vivo. ZVI-based NPs effectively downregulate the NRF2 gene both in vitro and in vivo, which suggested that ZVI-based NPs may provide a new option for NRF2 overexpressed chemoresistance in patients with serial cancers [114].

## 4. Conclusions and Perspectives

Herein, we detailed the recent progress made on the endogenous anti-cancer activity of ZVI-based NPs from the molecular, cellular, immunological, and TME points of view. The inherent anti-cancer activity of ZVI-based NPs was repetitively tested in vitro studies, using paired cancerous and control cells of lung, liver, and oral carcinomas, neuroblastoma, glioma, leukemia, cervical and breast adenocarcinomas. In in vivo study, 1–75 μg/kg was given through intravenous or intra-tumoral approaches in a single dosage or multiple dosages, and all of them demonstrated promising therapeutic outcomes without affecting the body weights of rodents. In spite most of the model cells being sensitive to the ZVI-based NPs treatment, a few cell lines are reported as ZVI-based NPs refractory cells, or only responded to a high dose of ZVI-based NPs [74]. This observation is accordant with the reported tumoral heterogenicity within cancers [115]. Intriguingly, it was proposed that such cell resistance can be sensitized with an extra ferroptosis inducer together with ZVI NPs [74]. The studies on ZVI-based NPs are still at the early stages and required future clinical translation. Considering the safety issues of ZVI-based NPs, it is of note that some metallic nanoparticles, including iron-based NPs, have been successfully approved by the U.S. Food and Drug Administration [116]. This brought up a new direction to achieve effective ferroptosis therapy with a lower dose of iron-based NPs combined with ferroptosis inducers. This promising development demands more comprehensive studies in the foreseeable future.

In conclusion, the high effectiveness and volatile bioactivity of ZVI-based NPs, compared to iron oxide nanoparticles, is attributed to NPs encountering the acidic TME as well as the active lysosomes within cancer cells. More specifically, the burst in release of iron ions from ZVI-based NPs increases the pro-inflammatory response and anti-cancer immune cell activity within the TME. At the subcellular level, unbalancing the molecular pathways between ROS and scavenging systems seems to be the main contributor to the observed specific cancer cell specific cell-death. Therefore, compared to conventional small compounds, e.g., doxorubicin, ZVI-based NPs serve as ‘iron ions-releasing bombs’ that can specifically activate within the acidic TME and in tumor cells. As a future outlook, one could take advantage of the magnetic properties of ZVI-based NPs, as they can also be combined with thermodynamic therapy approaches [117], magnetic resonance tumoral imaging [70,117,118,119], as well as computational tomography [120]. This is an enormous advantage allowing to tailor anti-cancer treatment regimes, including the monitoring of tumor localization and anti-cancer drug response, which will inherently benefit patient outcomes.

## Figures and Tables

**Figure 1 pharmaceutics-14-00099-f001:**
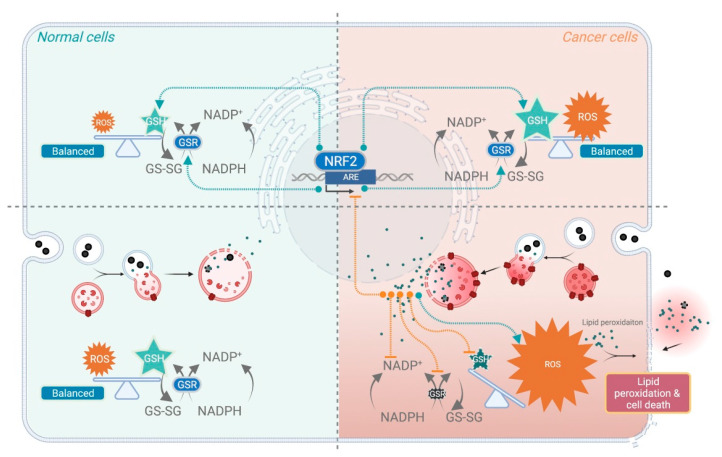
The mechanisms of ZVI-based NPs to selectively boost cancer cell death. Cancer cell specific induced cell death by ZVI-based NPs through shifting the balance between ROS and scavenging systems. Created with BioRender.

**Figure 2 pharmaceutics-14-00099-f002:**
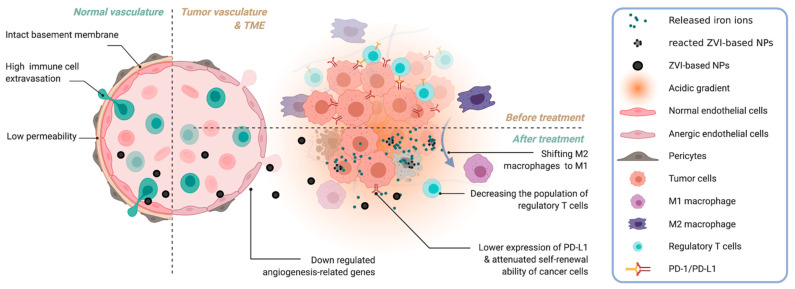
ZVI-NPs systemically deteriorate cancer cells through modulation of the surrounding blood vessels and immune cells that compose the TME. Adapted from “Features of Tumor Blood Vessels”, by BioRender (2021).

**Figure 3 pharmaceutics-14-00099-f003:**
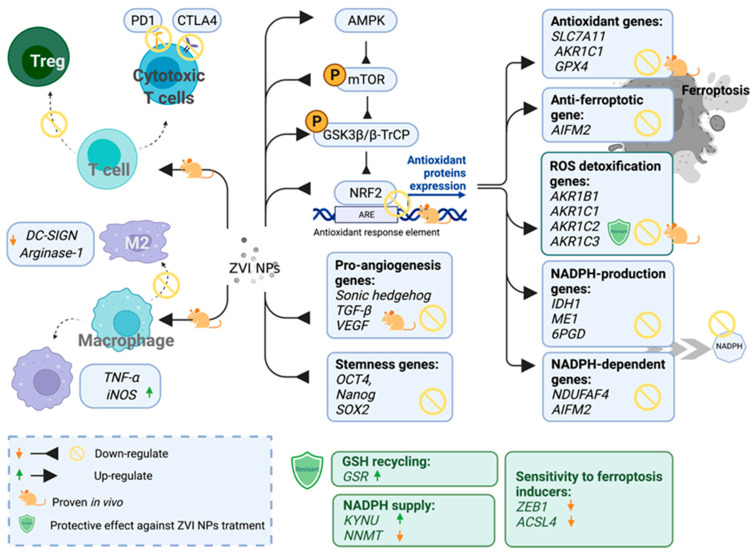
The overview of molecular modulation in vitro and in vivo in ZVI-based NP-treated cell and rodent models. There were several studies reporting genetic and molecular modulation under ZVI-based NPs treatment. The complex interaction and regulation of the different molecular networks involved are schematically depicted above. Created with BioRender.

**Table 1 pharmaceutics-14-00099-t001:** The anti-cancer activity of ZVI based NPs in vivo and in vitro.

Year	Authors	Materials	Sizes (nm)	In Vitro Study (IC50), Cell Types ^1^, Treating Periods	In Vivo Study
2021 [72]	Hsieh et al.	ZVI@Ag ZVI@CMC	81 ± 14 70 ± 14	About 10 μg/mL ZVI@Ag, A549 (human lung carcinoma), 48 h About 10 μg/mL ZVI@Ag, H460 (human lung carcinoma), 48 h About 5 μg/mL ZVI@Ag, H1299 (human lung carcinoma), 48 h ○ >> 50 μg/mL ZVI@Ag, MRC-5 (human normal lung fibroblasts), 48 h ○ >> 50 μg/mL ZVI@Ag, IMR-90 (human normal lung fibroblasts), 48 h About 10 μg/mL ZVI@CMC, A549 (human lung carcinoma), 48 h About 5 μg/mL ZVI@CMC, H460 (human lung carcinoma), 48 h About 1 μg/mL ZVI@CMC, H1299 (human lung carcinoma), 48 h About 5 μg/mL ZVI@CMC, LLC (human lung carcinoma), 48 h ○ >> 50 μg/mL ZVI@CMC, MRC-5 (human normal lung fibroblasts), 48 h ○ >> 50 μg/mL ZVI@CMC, IMR-90 (human normal lung fibroblasts), 48 h	25 mg ZVI@Ag or ZVI@CMC/kg intravenous, once a week for 4 weeks
2021 [67]	Liang et al.	Fe/Fe_3_O_4_ porous yolk shell NPs (PYSNPs)	15	21.9 μg(Fe)/mL, PYSNPs-2, Hep G2 (human liver carcinoma), 24 h 14.3 μg(Fe)/mL, iRGD-PYSNPs-2, Hep G2 (human liver carcinoma), 24 h 50–100 μg(Fe)/mL, iRGD-PYSNPs-2, L02 (normal liver cell), 24 h	1, 10 mg iRGD-PYSNPs /kg, intravenous
2020 [73]	Hashemi et al.	ZVI	10–30	2.5 μg/mL, AGS (human gastric adenocarcinoma), 24 h About 150 μg/mL, MCF-7 (human breast adenocarcinoma), 24 h ○ 37.5 μg/mL, BEAS-2B (human normal bronchial epithelial cells), 24 h	NA
2020 [68]	Yang et al.	ZVI@Ag	85 ± 17	1.0 μg/mL, OEC-M1 (human oral carcinoma), 48 h 6.1 μg/mL, DOK (human oral carcinoma), 48 h 0.9 μg/mL, OC3 (human oral carcinoma), 48 h ○ >>50 μg/mL, human oral epithelium cells, 48 h	40 mg/kg, intravenous (single injection)
2019 [69]	Yang et al.	ZVI@mSiO2	29 ± 7	5 μg/mL, OEC-M1 (human oral carcinoma), 24 h	40 mg/kg, intravenous
2019 [74]	Huang et al.	ZVI@CMC	50–100	0.6 μg/mL, OC3 (human oral carcinoma), 48 h 4.9 μg/mL, OEC-M1 (human oral carcinoma), 48 h 0.8 μg/mL, SCC9 (human oral carcinoma), 48 h >50 μg/mL, HSC-3 (human oral carcinoma), 48 h >50 μg/mL, SAS (human oral carcinoma), 48 h >50 μg/mL, KOSC-3 (human oral carcinoma), 48 h >50 μg/mL, OC-2 (human oral carcinoma), 48 h When combined treatment with ferroptosis inducer: 5 μg/mL, HSC-3 (human oral carcinoma), 48 h <5 μg/mL, SAS (human oral carcinoma), 48 h ○ >>5 μg/mL, human oral epithelium cells, 48 h	25 mg/kg, intravenous (4 injections)
2019 [76]	Anbouhi et al.	ZVI NPs	37	47.9 μg/mL, SH-SY5Y (human neuroblastoma), 24 h ○ Insignificant cytotoxicity at 100 μg/mL, white blood cells, 24 h	NA
2016 [70]	Shevtsov et al.	Fe(0)@MCM-41	250 × 150 with 3 nm pores	50–150 μg/mL, C6 (rat glioma), 24 h > 150 μg/mL, U87 (human glioblastoma), 24 h About 50 μg/mL, K562 (human leukemia), 24 h 50–150 μg/mL, Hela (human adenocarcinoma), 24 h ○ 50–150 μg/mL, Rat splenocytes, 24 h ○ About 50 μg/mL, Rat fibroblasts, 24 h	10 mg/kg, intravenous
2016 [77]	Zhang et al.	Amorphous iron NPs	10–15	About 100 μg/mL with 50 μM H_2_O_2_ at pH 6.5, MCF-7 (human breast adenocarcinoma), 24 h >> 200 μg/mL, without H_2_O_2_ or under pH 7.4	15 mg/kg, intratumor, 75 mg/kg, intravenous
2011 [43,71]	Wu et al.	Fe@Au	10–20	3.8 μg/mL, DOK (human oral carcinoma), 48 h 5.4 μg/mL, HCDB1 (hamster oral carcinoma), 48 h 2.1 μg/mL, SCC25 (human oral carcinoma), 48 h 0.6 μg/mL, OEC-M1 (human oral carcinoma), 48 h ○ >>50 μg/mL, human oral epithelium cells, 48 h	50 mg/kg, intratumor

^1^ ●: Cancerous cells; ○: normal non-cancerous cells.

## Data Availability

Not applicable.

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
