# Peer review of "From Microenvironment Remediation to Novel Anti-Cancer Strategy: The Emergence of Zero Valent Iron Nanoparticles"

_pharmaceutics, 2022, doi:10.3390/pharmaceutics14010099_

Round 1

Reviewer 1 Report

Authors have demonstrated the Novel Anti-Cancer Strategy: the Emergence of Zero Valent Iron Nanoparticles. I have just a few suggestions

1.The manuscript needs linguistic improvement.

2.Some citations and reference are missing.

Please add more background information about reactive oxydative species, so that it will bring the importance of antioxidant property of nanoparticles. ROS plays an important role in cancer development and ischemic stroke complications and so on.  (Please cite: 1. Chen et al. Semin Cancer Biol. 2020 Oct 6:S1044-579X(20)30203-0. doi: 10.1016/j.semcancer.2020.09.012.  2. Shekhar et al. International Journal of Molecular Sciences. 2021; 22(4):2074. https://doi.org/10.3390/ijms22042074)

Reviewer 2 Report

Wu and Yang et al. reviewed zero valent iron nanoparticles for novel Anti-Cancer treatment. It is a well-organized and comprehensive review. I recommend it for publication after the following points are addressed.

  1. Line 46-48, it is better to make the statement clearer that which treatment is using reactive oxygen species without selectivity.
  2. Line 57-59, several recent studies (doi.org/10.1021/acs.langmuir.8b00851; doi.org/10.1021/acs.biomac.0c01704) are recommended to be included to support such claim.
  3. In the section of 2.2, the authors should discuss about the coating of NPs in details. In the current form of this ms., most of the coatings are inorganic materials, which can’t be stable or biocompatible in the biological mediums or fluids.

Reviewer 3 Report

The authors provide an extensive and detailed review on the role of Zero Valent Iron Nanoparticles as novel anti-cancer agents. The manuscript it is well well organized a cover all the aspect of this new field.

My only comment is that in the introduction part, a short section should be implemented to give to the reader an overview of different advanced anticancer approaches that are currently investigate (for exemple ACS Applied Materials & Interfaces 13 (11), 12997-13008 and 10.1016/j.ejpb.2015.03.018)

)

Round 2

Reviewer 1 Report

strongly suggest to publish.